# DIFF-TRANSFER: MODEL-BASED ROBOTIC MANIPULATION SKILL TRANSFER VIA DIFFERENTIABLE PHYSICS SIMULATION

## ABSTRACT

The capability to transfer mastered skills to accomplish a range of similar yet novel tasks is crucial for intelligent robots. In this work, we introduce *Diff-Transfer*, a novel framework leveraging differentiable physics simulation to efficiently transfer robotic skills. Specifically, *Diff-Transfer* discovers a feasible path within the task space that brings the source task to the target task. At each pair of adjacent points along this task path, which is two sub-tasks, *Diff-Transfer* adapts known actions from one sub-task to tackle the other sub-task successfully. The adaptation is guided by the gradient information from differentiable physics simulations. We propose a novel path-planning method to generate sub-tasks, leveraging $Q$-learning with a task-level state and reward. We implement our framework in simulation experiments and execute four challenging transfer tasks on robotic manipulation, demonstrating the efficacy of Diff-Transfer through comprehensive experiments. Supplementary and Videos are on the website https://sites.google.com/view/difftransfer

## 1 INTRODUCTION

The capacity for rapidly acquiring new skills in object manipulation is crucial for intelligent robots operating in real-world environments. One might wonder, how can robots efficiently learn manipulation skills across diverse objects? A straightforward approach would involve teaching a robot a new manipulation skill for every distinct object and task. However, this method lacks efficiency and is infeasible due to the vast variety of objects and possible robot interactions. Nonetheless, we could also notice that different manipulation skills may share common properties. As shown in Fig. 1, the one-directional pushing skill could be correlated to an object reorientation skill. Thus, it may be feasible to leverage prior knowledge acquired from one task to aid in learning another similar task. Transferring this prior knowledge and acquired skill set to new tasks could greatly enhance learning efficiency compared to starting from scratch.

Our intuition to solve this transfer learning problem is that Newton's Laws apply universally in our physical world. Therefore, when involved in similar tasks where objects are moved by similar poses, robots should interact with objects in similar ways. In this way, efficiently leveraging the local information hidden in the variation of manipulation tasks could be the key to efficient task transfer learning.

In this paper, we investigate the problem of transferring manipulation skills between two object manipulation tasks. Our proposed framework is depicted in Fig. 1. We approach this problem by interpolating the source task and target task by producing a large number of intermediate sub-tasks between them which gradually transform from the source task toward the target task. These continuously and gradually transforming intermediate sub-tasks act as the bridge for transferring the action sequence from the source task to the target task.

To better leverage the physical property associated with the object shape and pose transformation, we leverage differentiable simulation to capture model-based gradient information and use it in transforming robot action sequences. We introduce a refined $Q$-learning method for path planning in the pose transfer problem, where we use a high-level state and a well-designed reward to generate the path of seamlessly connected sub-tasks with a sample-based searching method.

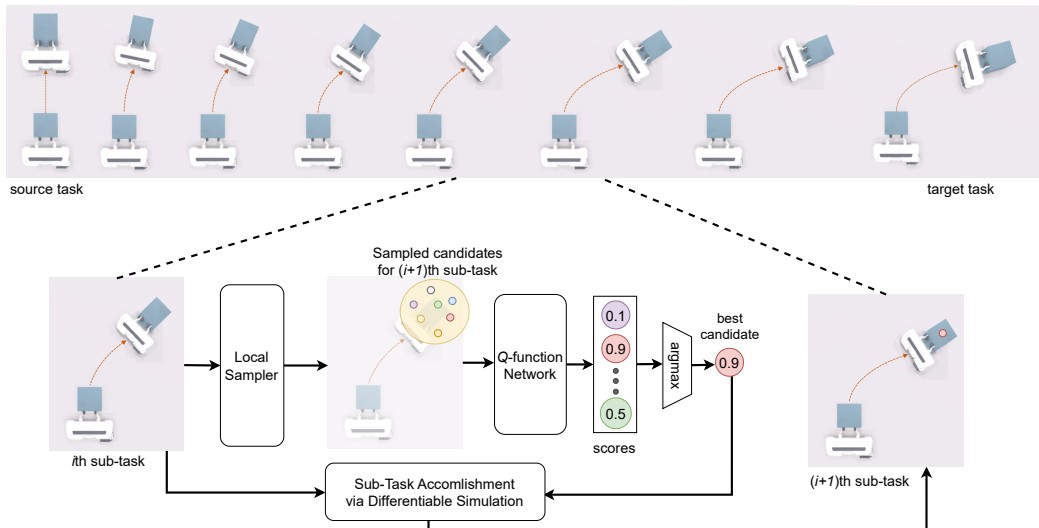

Figure 1: The overall approach of *Diff-Transfer* includes a path of $L - 1$ sub-tasks. *Diff-Transfer* leverages *Local Sampler*, *Q-function Network* and *argmax* function to select the best candidate to generate the $(i+1)$th sub-task given the $i$th sub-task, and learn the action sequence via differentiable physics simulation.

We execute a series of challenging manipulation tasks using Jade(Yang et al., 2023), a differentiable physics simulator designed for articulated rigid bodies. We undertake four tasks: *Close Grill*, *Change Clock*, *Open Door*, and *Open Drawer*. The outcomes demonstrate that our system surpasses prevalent baselines for transfer learning and direct transfer without path planning through differentiable simulation, highlighting the efficacy and merits of our approach. Additionally, we perform several ablation studies.

In summary, we make the following contributions:

- We propose a systematic framework for model-based transfer learning, leveraging the differentiable physics-based simulation and applying our framework for pose transfer and object shape transfer.
- We propose a novel path planning method for generating multiple sub-tasks in the task space and learning an action sequence for a new sub-task with the proximity property and leveraging $Q$-learning and differentiable physics simulation.
- We conduct comprehensive experiments to demonstrate the effectiveness of our proposed transfer learning framework.

## 2 RELATED WORK

### 2.1 DIFFERENTIABLE SIMULATION FOR MANIPULATION.

Significant advancements have been achieved in the field of differentiable physics engines, thanks to the evolution of automatic differentiation techniques (Paszke et al., 2019; Team et al., 2016; Hu et al., 2019a; Bell, 2020; Bradbury et al., 2018; Agarwal et al.). Various differentiable physics simulations have been developed for specific applications, such as rigid bodies (de Avila Belbute-Peres et al., 2018; Degrave et al., 2019; Yang et al., 2023), soft bodies (Hu et al., 2019a;b; Jatavallabhula et al., 2021; Geilinger et al., 2020; Du et al., 2021), cloth (Liang et al., 2019; Qiao et al., 2020; Li et al., 2022; Yu et al., 2023), articulated bodies (Werling et al., 2021; Ha et al., 2017; Qiao et al., 2021), and fluids (Um et al., 2020; Wandel et al., 2020; Holl et al., 2020; Takahashi et al., 2021). Several studies have applied differentiable physics simulations to robotic manipulations. Turpin et al. (2022) focused on multi-fingered grasp synthesis, while Lv et al. (2022) guided robots in manipulating articulated objects. Zhu et al. (2023a;b) enabled model-based learning from demonstrations

by optimizing over dynamics, and Lin et al. (2022a;b) targeted deformable object manipulation. Yang et al. (2023) developed a differentiable simulation called *Jade* for articulated rigid bodies with Intersection-Free Frictional Contact.

However, the incorporation of contact dynamics often results in non-convex optimization challenges due to discontinuities from contact mode switching (Suh et al., 2022; Antonova et al., 2022; Zhu et al., 2023a). To mitigate this, contact-centric trajectory planning has been proposed (Mordatch et al., 2012; Marcucci et al., 2017; Cheng et al., 2021; Gabiccini et al., 2018; Zhu et al., 2023a; Chen et al., 2021), which plans both contact points and forces and generate manipulation actions afterward. Additionally, Pang et al. (2022) introduced smoothing techniques for contact gradients and employed a convex quasi-dynamics model for feasible action searching. In alignment with existing research, our study utilizes differentiable physics simulations for the purpose of transferring robotic manipulation skills across different task spaces, thereby facilitating model-based transfer learning.

## 2.2 TRANSFER LEARNING IN ROBOTICS.

Transfer learning has become a cornerstone in robotics, aiming to generalize skills across varying tasks, environments, or robotic platforms. Although still an open challenge, the majority of research has employed reinforcement learning for skill transfer (Taylor & Stone, 2009). Several approaches have been proposed to address this challenge. Lazaric et al. (2008); Xu et al. (2021); Jian et al. (2021) utilize domain randomization during training to enhance agent robustness across diverse physical environments and to focus on task-relevant features. Tirinzoni et al. (2018); Hu et al. (2023) fine-tune reward and value functions on new tasks, while Konidaris & Barto (2007); Liu et al. (2021); Zhao et al. (2022) directly adapt policies to new environments. Finn et al. (2017) introduces a meta-learning framework to improve agent adaptability across various tasks. Chi et al. (2022) employs an iterative policy and approximates residual dynamics for runtime adaptation. Distinct from these approaches, our work adopts a model-based perspective for policy transfer. We utilize differentiable simulations to approximate physical dynamics and directly optimize pre-existing policies. We address the key differences between source and target environments as rewards where we accommodate varying manipulation goals that yield different reward functions.

## 3 PROBLEM STATEMENT

We consider two object manipulation tasks on a robot with $m$ joints. We assume the source manipulation task is specified by the goal of object pose change $\Delta s_{\text{source}} \in \mathbb{R}^6$. Suppose applying a given expert action sequence $A_{\text{source}} = [a_{\text{source}}^{(t)}]_{t=1}^T$ on the task would yield a state-action trajectory $\tau_{\text{source}} = [(s_{r,\text{source}}^{(t)}, s_{o,\text{source}}^{(t)}, a_{\text{source}}^{(t)})]_{t=1}^T$ where $s_{r,\text{source}}^{(t)} \in \mathbb{R}^m$, $s_{o,\text{source}}^{(t)} \in \mathbb{R}^6$, $a_{\text{source}}^{(t)} \in \mathbb{R}^m$ denotes robot state, object state and robot action at time $t$. We assume action sequence $A_{\text{source}}$ can successfully complete the task, i.e. moving the object from the starting pose $s_{o,\text{source}}^{(1)}$ to the goal pose $s_{o,\text{source}}^{(T)} = s_{o,\text{source}}^{(1)} + \Delta s_{\text{source}}$. Our objective is to derive an action sequence $A_{\text{target}} = [a_{\text{target}}^{(t)}]_{t=1}^T$ that can successfully complete a new target manipulation task $\Delta s_{\text{target}}$ specified by the goal of object pose change $\Delta s_{\text{target}}$.

## 4 TECHNICAL APPROACH

We approach this problem by defining a path consisting of $L$ tasks

$$\mathcal{P} = [\Delta s_1, \Delta s_2, \ldots, \Delta s_L] \tag{1}$$

that connects the source and target tasks where $\Delta s_1 = \Delta s_{\text{source}}$ is the source task and $\Delta s_L = \Delta s_{\text{target}}$ is the target task. Our approach consists of $L - 1$ steps of action transfer. At step $i$, our goal is to transfer a well-optimized action sequence $A_i$ on task $\Delta s_i$ to be a well-optimized action sequence $A_{i+1}$ on the next task in the sequence $\Delta s_{i+1}$. For any $i$, we assume the difference between tasks $\Delta s_i$ and $\Delta s_{i+1}$ is sufficiently small so that the it is relatively easy to use local information such as differentiable simulation gradient to optimization for actions transfer.

$$||\Delta s_i - \Delta s_{i+1}|| < \varepsilon_1 \tag{2}$$

where $\varepsilon_1$ denotes the upper limit between the final object state for two consecutive sub-tasks. This property is crucial to our gradient-based method in the following sub-section.

## 4.1 How to accomplish a sub-task

Our approach to deduce the requisite actions is through a gradient-based methodology. Under the assumption that the subsequent sub-task goal pose deviates from the current goal pose with a limited distance as described in Eq. 2, we posit that the actions for the sub-task are in close proximity to the actions of the source. This postulation naturally lends itself to the application of gradient descent for optimization. We aim to optimize our current action sequence $\{a_{\text{cur}}^{(t)}\}_{t=1}^{T}$, denoted as $A_{\text{cur}}$, with its initialization of $A_i$. The rollout trajectory based on $A_{\text{cur}}$ is denoted $\tau_{\text{cur}} = \{(s_{r,\text{cur}}^{(t)}, s_{o,\text{cur}}^{(t)}, a_{\text{cur}}^{(t)})\}_{t=1}^{T}$

To elaborate, for each specific task, we introduce a loss function, $\mathcal{L}_{task}$.

$$\mathcal{L}_{task} = ||\Delta s_{\text{cur}} - \Delta s_{i+1}||^2 \tag{3}$$

where $\Delta s_{\text{target}}$ is the object pose change of $(i + 1)$th sub-task goal and $\Delta s_{\text{cur}}$ is the object pose change of our rollout trajectory. We regard the task as accomplished if $\mathcal{L}_{task}$ is smaller than a certain threshold $\varepsilon_t$.

Utilizing the capabilities of the differentiable simulation framework Jade, we compute the gradient $\left\{\frac{\partial \mathcal{L}_{task}}{\partial a_{\text{cur}}^{(t)}}\right\}_{t=1}^{T}$, denoted as $\frac{\partial \mathcal{L}_{task}}{\partial A_{\text{cur}}}$. Subsequently, the current actions $A_{\text{cur}}$ are updated to minimize the task loss $\mathcal{L}_{task}$.

$$A_{\text{cur}} \leftarrow A_{\text{cur}} - \eta \frac{\partial \mathcal{L}_{task}}{\partial A_{\text{cur}}} \tag{4}$$

Thus we introduce Algorithm 1 as a function TRANSFERSTEP, since we will reuse this function in Section 4.1. It takes the trajectory $\tau_i$ for $i$th sub-task and the object pose change $\Delta s_{i+1}$ for $(i+1)$th sub-task as input. And it will output the optimized task loss $\mathcal{L}_{task}$, the boolean value $X$ indicating if the sub-task is successfully completed, and the rollout trajectory $\tau_{i+1}$ based on the optimized actions $A_{\text{cur}}$. If $X$ is True, then $A_{\text{cur}}$ is the desired $A_{i+1}$. This algorithm iteratively refines the action sequence $A_{\text{cur}}$ over a maximum of $n_{epoch}$ iterations or until a convergence criterion is met.

---

**Algorithm 1** Sub-Task Accomplishment

---

1: **Input:** $\tau_i = \{(s_{r,i}^{(t)}, s_{o,i}^{(t)}, a_i^{(t)})\}_{t=1}^{T}, \Delta s_{i+1}$
2: **Output:** $\mathcal{L}_{task}, X, \tau_{i+1}$
3: **function** TRANSFERSTEP($\tau_s, \Delta s_{i+1}$)
4:      $s_{r,\text{cur}}^{(1)} \leftarrow s_{r,i}^{(1)}, a_{\text{cur}}^{(t)} \leftarrow a_i^{(t)}, t = 1, 2, \ldots, T$
5:      **for** $e$ **in** $1, 2, \ldots, n_{epoch}$ **do**
6:          **for** $t$ **in** $1, 2, \ldots, T - 1$ **do**
7:              $(s_{r,\text{cur}}^{(t+1)}, s_{o,\text{cur}}^{(t+1)}) \leftarrow$ **simulate**$(s_{r,\text{cur}}^{(t)}, s_{o,\text{cur}}^{(t)}, a_{\text{cur}}^{(t)})$
8:          $\Delta s_{\text{cur}} \leftarrow s_{o,\text{cur}}^{(T)} - s_{o,\text{cur}}^{(1)}$
9:          $\mathcal{L}_{task} \leftarrow ||\Delta s_{\text{cur}} - \Delta s_{i+1}||^2$
10:          $A_{\text{cur}} \leftarrow A_{\text{cur}} - \eta \frac{\partial \mathcal{L}_{task}}{\partial A_{\text{cur}}}$
11:          **if** $\mathcal{L}_{task} \leq \varepsilon_t$ **then**
12:              **return** $\mathcal{L}_{task}$, **True**, $\{(s_{r,\text{cur}}^{(t)}, s_{o,\text{cur}}^{(t)}, a_{\text{cur}}^{(t)})\}_{t=1}^{T}$
13:      **return** $\mathcal{L}_{task}$, **False**, $\{(s_{r,\text{cur}}^{(t)}, s_{o,\text{cur}}^{(t)}, a_{\text{cur}}^{(t)})\}_{t=1}^{T}$

---

## 4.2 Sub-Tasks Generation

Given Algorithm 1 and the path $\mathcal{P}$, it is easy to compute the optimized actions $A_t$ for our target task, since we can use dynamic programming to optimize $A_{i+1}$ based on $A_i$. The only problem is

to generate one feasible path $\mathcal{P}$ where not only the property in Eq. 2 holds but also the Algorithm 1 tends to return the successful result with optimized action sequence $A_{i+1}$ and the corresponding trajectory $\tau_{i+1}$ for $(i+1)$th sub-task for each index $i$. This reduces the problem into a path planning problem in the goal pose space where each node in the space denotes a goal final object state and we aim to build a path connecting the source goal pose and the target one.

While there are lots of traditional path-planning algorithms in 3-D Euclidean space, they fail to solve our problem because the goal pose space is in a higher dimension and the obstacle is harder to detect. We introduce our innovative reinforcement learning method by predicting the difficulty of sub-tasks using a refined $Q$-function neural network $Q(x; \theta)$ parameterized by $\theta$. Instead of taking input of the conventional state and action at time $t$, the network takes a high-level state input $x$, which could be any object pose change like $\Delta s_{\text{target}}$. The output $r$ would be the estimated reward.

Unlike traditional RL problems with clear task rewards, the reward in our problem needs an elaborate design because we are performing path planning on a higher task-space level. We introduce the reward function as

$$r(x) = -(\lambda_t \cdot \mathcal{L}_{task} + \lambda_d \cdot ||x - \Delta s_{\text{target}}||^2) \tag{5}$$

To illustrate this equation, the first term $\mathcal{L}_{task}$ is computed using Eq. 3 where $\Delta s_{i+1}$ is given as $x$ and $\Delta s_{\text{cur}}$ is given by the optimized actions $A_{\text{cur}}$ for sub-task goal $x$. The second term $||x - \Delta s_{\text{target}}||^2$, shortly as $\mathcal{L}_{dis}$, describes the distance from pose change $x$ to the target pose change $\Delta s_{\text{target}}$. Finally, $\lambda_t$ and $\lambda_d$ are weight coefficients to balance these two terms. Therefore, such reward results in a better path-planning algorithm because when the reward is high, both the task loss $\mathcal{L}_{task}$ and the distance to target goal $\mathcal{L}_{dis}$ are low.

Suppose we have the accurate $Q(x; \theta)$ network, we can generate the path $\mathcal{P}$ in either a gradient-based way or a sample-based way. We employ the sampled-based approach for the current pose transfer problem to increase the robustness of stochastic noise from the inaccurate network in reality. In detail, given $i$th sub-task with a pose change $\Delta s_i$, we sample $n$ vectors $\{x_j\}_{j=1}^n$, denoted as $S$, in the task space in the neighbourhood of the $i$th sub-task goal $\Delta s_i$, so that

$$||\Delta s_i - x_j|| < \varepsilon_{sample}, j = 1, 2, \ldots, n \tag{6}$$

where $\varepsilon_{sample}$ is the radius of the neighbourhood. In these $n$ candidates for the $(i+1)$ sub-task, we choose the best one $k$ based on our current knowledge to maximize the reward $r_k$

$$k = \arg \max_j r_j, j = 1, 2, \ldots, n \tag{7}$$

Once we get the best candidate $x_k$, we call the function TRANSFERSTEP in Algorithm 1, in an attempt to optimize an action sequence $A_{i+1}$ for the given $(i+1)$th sub-task. Should this process be successful, we shall continue to generate the next sub-task recursively until the target goal is attained. Otherwise, we shall discard this candidate $x_k$ and find an alternative best candidate from $S$ iteratively, as is shown in Algorithm 2.

To learn an approximate network $Q(x; \theta)$, we maintain a dataset $D$ dynamically during the path-planning process. Each time after we call the TRANSFERSTEP function and get more information about the task space, we add the data pair $(x_k, r_k)$ into $D$, update $\theta$ with the $Q$-learning method to gain a better network and proceed on path planning.

### 4.3 IMPLEMENTATION DETAILS

In this section, we discuss the implementation details of *Diff-Transfer* in Algorithm 2. To begin with, we pre-train our network $Q(x; \theta)$ with a refined initial reward in Eq. 5, where $\mathcal{L}_{task}$ is set to a certain constant $c_t$ because we cannot know the difficulty of any sub-task beforehand. Specifically, we generate labels $(x_{\text{pre}}, r_{\text{pre}})$ randomly to build a dataset $D_{\text{pre}}$ and use it to fit $\theta$ using a supervised learning method via minimizing the loss $l_{\text{pre}}(\theta) = ||Q(x_{\text{pre}}; \theta) - r_{\text{pre}}||^2$. With online dataset $D = \{(x_k, r_k)\}_{k=1}^m$ collected during execution of our path-planning method, network parameters $\theta$ will be fine-tuned to minimize the loss $l(\theta) = ||Q(x_k; \theta) - r_k||^2$. It is worth noting that $D$ doesn't contain

data from $D_{\text{pre}}$ because data in $D$ collected from rollouts in simulation reflect the actual rewards of sub-tasks while $D_{\text{pre}}$ just provides a rough estimation under the hypothesis that all sub-tasks have same difficulties, which is hardly true in the real transfer problem.

---

**Algorithm 2** $Q$-function Network Guided Path Planning

---

1: **function** PATHSEARCH($\tau_i, \Delta s_{\text{target}}$)
2:     **if** $\|\Delta s_i - \Delta s_{\text{target}}\| \leq \varepsilon_{pose}$ **then**
3:         **return** $\tau_i$
4:     Randomly sample $n$ vectors $S \leftarrow \{x_j\}_{j=1}^n$ in the neighbourhood of $\Delta s_i$
5:     $r_j \leftarrow Q_\theta(x_j), j = 1, 2...n.$
6:     **while** $S \neq \varnothing$ **do**
7:         $k \leftarrow \arg\max_j r_j$
8:         $\mathcal{L}_{task}, X, \tau_{i+1} \leftarrow$ TRANSFERSTEP($\tau_i, x_k$)
9:         $\mathcal{L}_{dis} \leftarrow \|x_k - \Delta s_{\text{target}}\|^2$
10:        $r_k \leftarrow -(\lambda_t \cdot \mathcal{L}_{task} + \lambda_d \cdot \mathcal{L}_{dis})$
11:        $D \leftarrow D \cup \{(x_k, r_k)\}$
12:        Update $\theta$ using dataset $D$
13:        **if** $X = $ **True then**
14:            PATHSEARCH($\tau_{i+1}, \Delta s_{\text{target}}$)
15:        **else**
16:            $S \leftarrow S - \{x_k\}$
17:            **continue**
18:     **return** failure

---

## 5   EXPERIMENTS

In this section, we present a rigorous experimental framework meticulously designed to elucidate the effectiveness of our proposed system *Diff-Transfer*. This exhaustive evaluation encompasses an assessment of the system's performance across diverse conditions, while also subjecting it to rigorous scrutiny in the presence of unforeseen challenges. The tests conducted in this study are geared towards offering a comprehensive panorama of the system's capabilities. Our foremost objective is to substantiate the theoretical foundations expounded earlier and establish a seamless connection between theory and practical implementation, thereby affirming the system's scalability and adaptability across a multitude of application domains.

### 5.1   EXPERIMENTAL SETUP

#### 5.1.1   SIMULATION SETTING

We choose multiple manipulation tasks from RLBench (James et al., 2020) and adapt the environment to the Jade(Yang et al., 2023) simulation. Specifically, we acquire the trajectory of states for each task, along with the objects' Unified Robot Description Format (URDF) files and corresponding mesh files. Actions are computed utilizing inverse dynamics and optimization within Jade, providing us with a comprehensive initial trajectory of both states and actions, denoted as $\tau_{\text{source}}$.

#### 5.1.2   EVALUATION METRIC

We employ the number of iterations $N$ in the optimization loop to evaluate the efficiency of our methods and compare the results. We also report the distance $d$, which is a task-related metric describing the completeness of manipulation. For each specific manipulation task, we run 5 times our method to reduce the effect of randomness and report the mean value for both the iterative steps and the distance as $\bar{N}$ and $\bar{d}$, and the standard deviation as $\sigma_N$ and $\sigma_d$.

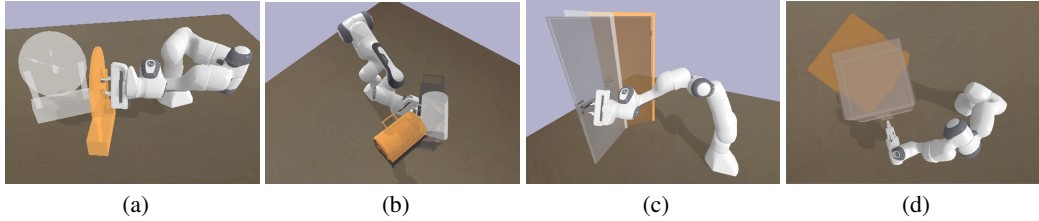

|     (a)     |     (b)     |     (c)     |     (d)     |

Figure 2: Source Task(grey object) and Target Task(orange object) for (a) *Change Clock*, (b) *Close Grill*, (c) *Open Door*, and (d) *Open Drawer*.

| Method | *Diff-Transfer* | | | | *MAML* | | *DMG* | | *Direct Transfer* | | |
|---|---|---|---|---|---|---|---|---|---|---|---|
| Task Name | $\bar{N}$ | $\sigma_N$ | $\bar{d}$ | $\sigma_d$ | $d$ | success | $d$ | success | $N$ | $d$ | success |
| *Change Clock* | **55.6** | 61.1 | **3.72** | 1.38 | 10.27 | × | 27.46 | × | 1000+ | 19.66 | × |
| *Close Grill* | **66.4** | 11.5 | **1.80** | 0.55 | 18.54 | × | 56.71 | × | 1000+ | 8.53 | × |
| *Open Door* | **57.8** | 38.2 | **0.64** | 0.43 | 9.20 | × | 41.91 | × | 255 | 1.40 | ✓ |
| *Open Drawer* | **123.8** | 103.9 | **0.06** | 0.00 | 0.08 | × | 0.18 | × | 1000+ | 0.12 | × |

Table 1: Experiment Results for *Diff-Transfer*, *MAML*, *DMG*, and *Direct Transfer*. *Diff-Transfer* is executed using 5 distinct random seeds.

### 5.1.3 MANIPULATION SKILL TRANSFER TASKS

***Close Grill***   The robot is required to close a grill lid. This task is considered successful if the grill lid has been rotated to close. The distance $d$ describes the distance from the final angle of the grill lid joint to the target angle, with a unit of degrees.

***Change Clock***   The robot is required to change a clock. This task is considered successful if the clock pointer has been revoluted to a specific orientation. The distance $d$ describes the distance from the final angle of the clock pointer to the target angle, with a unit of degrees.

***Open Door***   The robot is required to open a door. This task is considered successful if the door has been rotated to a specific orientation from the door frame. The distance $d$ describes the distance from the final angle of the door to the target angle, with a unit of degrees.

***Open Drawer***   The robot is required to open a drawer. The chest has 3 drawers. This task is considered successful if the specific drawer has been pulled out from the chest. The distance $d$ describes the distance from the final translation of the drawer to the target angle, with a unit of meters.

### 5.1.4 IMPLEMENTATION DETAILS

To illustrate the details presented in Section 4, we define $\Delta s_i$, the objective of the $i$th sub-task, as the base pose change of the manipulated object from its pose in the source task. This definition slightly diverges from the description in Section 3, as these intricate manipulation tasks require the robot to manipulate the object's joint, rather than altering its pose by pushing.

We employ a three-layer MLP to implement the Q-function network $Q(x; \theta)$. Rather than directly utilizing the reward function in Eq. 5, we characterize the output network as an estimated loss with a value of $-r(x)$, explaining why the landscapes in Fig. 3 exhibit a minimum area instead of a maximum, a point to be discussed in subsequent Section 5.3.

## 5.2 BASELINE

**DMP**   DMP (Dynamic Movement Primitives) is a method for learning and reproducing complex dynamic movement skills in robots and other systems, making it easier for them to perform tasks

| Method | _Diff-Transfer_ | | | | _Diff-Transfer ($\lambda_t = 0$)_ | | | | _Linear Interpolation_ | | |
|---|---|---|---|---|---|---|---|---|---|---|---|
| Task Name | $\bar{N}$ | $\sigma_N$ | $\bar{d}$ | $\sigma_d$ | $\bar{N}$ | $\sigma_N$ | $\bar{d}$ | $\sigma_d$ | $N$ | success | $d$ |
| Change Clock | 55.6 | 61.1 | 3.72 | 1.38 | **51.0** | 28.7 | **3.23** | 1.70 | 68.0 | ✓ | 5.43 |
| Close grill | **66.4** | 11.5 | **1.80** | 0.55 | 96.6 | 28.4 | 2.45 | 0.55 | 157.0 | ✓ | 3.36 |
| Open Door | **57.8** | 38.2 | **0.64** | 0.43 | 185.4 | 118.3 | 2.78 | 2.16 | 113.0 | ✓ | 4.11 |
| Open Drawer | **123.8** | 103.9 | **0.06** | 0.00 | 527.0 | 712.0 | **0.06** | 0.00 | 309.0 | ✗ | 0.38 |

Table 2: Experiment Results for _Diff-Transfer_, _Diff-Transfer ($\lambda_t = 0$)_, and _Linear Interpolation_. Both _Diff-Transfer_ and _Diff-Transfer ($\lambda_t = 0$)_ are executed using 5 distinct random seeds.

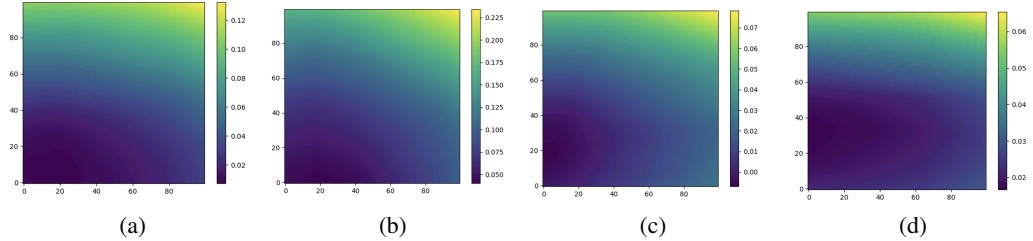

|(a)|(b)|(c)|(d)|

Figure 3: Visualization of learned $Q$-function Landscapes for (a) _Change Clock_, (b) _Close Grill_, (c) _Open Door_, and (d) _Open Drawer_. The $x$-axis represents translation, and the $y$-axis represents orientation. The origin symbolizes the change in target pose, $\Delta s_{\text{target}}$, while the top right corner denotes the change in source task pose, $\Delta s_{\text{source}}$.

such as reaching and grasping objects. Specifically, for a transfer task, we use the robot trajectory of the source task to fit the dmp function, modify the object target on the target task and reproduce the motion trajectory.

**MAML** Model-agnostic meta-learning (MAML) is a meta-learning algorithm that enables machine learning models to quickly adapt to new tasks with minimal training data by learning good initializations that can be fine-tuned for specific tasks, making it highly applicable to a variety of applications. application. Specifically, for a transfer task, we perform learning on 4 source tasks and perform trajectory prediction on a target task. In our experiments, the trained policy is a two-layer MLP network with 128 hidden units in each layer. We use the adam optimizer and SGD loss function to train the policy for 1000 epochs. In each epoch, we perform task-level training and meta-training. During each task-level training, we sample 20 trajectories on four source tasks to update the parameters of the task-level strategy. During each meta-training, we use task-level update parameters to sample 5 trajectories on 4 source tasks and update the policy parameters. We will train the final trained policy on the target task for 20 epochs to fine-tune the parameters, and calculate whether the policy given at this time can complete the target task.

**Direct Transfer** To demonstrate the efficacy of our path-searching method, we assess the direct transferring technique on each task, using it as one of the baselines, denoted as _Direct Transfer_. Contrary to constructing a path where the source task and the target task are cohesively linked via several intermediate sub-tasks as in Algorithm 2, _Direct Transfer_ solely endeavors to optimize an action sequence for the target task, directly drawing from the source task trajectory through differentiable simulation, as outlined in Algorithm 1.

## 5.3 EXPERIMENT RESULTS

The iteration counts $N$ and distances $d$ are detailed in Table 1 for _Diff-Transfer_, _MAML_, _DMG_, and _Direct Transfer_. As illustrated in the table, our algorithm manifests superior efficacy across all evaluated tasks. While _MAML_ and _DMG_ are unable to successfully accomplish any of the four tasks, and _Direct Transfer_ only yields a successful outcome in the _Open Door_ task, our _Diff-Transfer_ manages to fulfill all four tasks, achieving a success rate of 100% across 5 varied random seeds. Additionally,

*Diff-Transfer* requires significantly fewer iterative steps compared to *Direct Transfer* to accomplish the transfer task, underscoring the criticality of constructing a seamless path to mitigate the complexity of each sub-task transfer, and highlighting that attempts to transfer via brute force are frequently either impractical or necessitate more iterations. Regarding *MAML* and *DMG*, these methods, being somewhat antiquated, struggle to finalize this innovative transfer task within a reasonable time.

To confirm the validity of our path-planning approach, we have depicted the landscape of our $Q$-function network in Fig. 3. In each depiction, the horizontal axis denotes the translation, and the vertical axis denotes the orientation, together constituting a task space for any alterations in pose. The origin represents the target pose change $\Delta s_{\text{target}}$ while the top right corner represents the source task pose change $\Delta s_{\text{source}}$. As exhibited in the images, there exists a minimum area surrounding the origin, indicating that the network directs correctly toward the target task. Moreover, this area does not necessarily need to be precisely at the origin; given the varying complexities of different tasks, completing a sub-task pose near the $\Delta s_{\text{source}}$ is often more feasible, resulting in a lower value of $\mathcal{L}_{task}$ in Eq. 3 and, subsequently, contributing to a reduced total loss. This task-level characteristic elucidates why these landscapes exhibit a similar pattern with the aforementioned minimum area around the origin, aligning with our anticipations, even though the low-level manipulations might significantly diverge.

### 5.4 ABLATION STUDY: *Employ Different Path-Planning Methods*

We conduct two different ablation tests for *Diff-Transfer* with distinct path-planning methods.

1. We remove the Q-learning network and replace it with a deterministic linear interpolation method between $\Delta s_{\text{source}}$ and $\Delta s_{\text{target}}$, denoted as *Linear Interpolation*.

2. We refine the reward function in Eq. 5 by removing the task loss term, with $\lambda_t = 0$, denoted as *Diff-Transfer ($\lambda_t = 0$)*.

Our experiment results for the ablation study are presented in Table 2. Generally speaking, both *Diff-Transfer* and *Diff-Transfer ($\lambda_t = 0$)* achieve a $100\%$ success rate across four tasks, employing $5$ distinct random seeds, while Linear Interpolation succeeds in three out of the four transfer tasks. This denotes that path planning, even by naive methods, can substantially elevate the success rate in transferring manipulation tasks. To elaborate, the data reveals that our *Diff-Transfer* excels in tasks such as *Close grill*, *Open Door*, and *Open Drawer*, exhibiting quicker convergence (smaller $N$) and heightened precision in manipulation outcomes (smaller $d$) compared to *Diff-Transfer ($\lambda_t = 0$)* and *Linear Interpolation*. Regarding the *Change Clock* task, *Diff-Transfer*, *ablation*, and *Linear Interpolation* display comparable performance, suggesting that accomplishing this transfer task via differentiable physics simulation is relatively uncomplicated. In conclusion, the path-planning methodology employed in *Diff-Transfer* is imperative and efficient, leading to enhanced success rates and reduced time expenditures in most instances.

## 6 CONCLUSION

In this paper, we introduced an advanced framework aiming to revolutionize the paradigm of robotic manipulation skill acquisition through transfer learning. Drawing inspiration from the omnipresence of Newtonian principles, our method centers on the potential to generalize manipulation strategies across object poses in 3-D Euclidean space. To navigate the complex landscape, we instigate a bridge mechanism, employing a continuum of intermediate sub-tasks as conduits for the seamless relay of skills between distinct object poses, where the path of sub-tasks is generated through a refined $Q$-function network with task-level states and rewards. This focus is further bolstered by our integration of differentiable simulation, affording us an intricate understanding of the physical intricacies inherent in pose transformations. The compelling results from our meticulous experiments underscore the robustness and efficacy of our proposed framework. In summation, our pioneering contributions herald a new era in robotic adaptability, reducing the dependency on ground-up learning and accelerating the skill transfer processes, particularly in the realms of manipulations with different object poses.

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
