# OpenReview forum: "Diff-Transfer: Model-based Robotic Manipulation Skill Transfer via Differentiable Physics Simulation"
_ICLR.cc/2024/Conference — Submitted to ICLR 2024_

### Official Review · Reviewer_PT7k · 2023-10-30

**Soundness:** 2 fair
**Presentation:** 3 good
**Contribution:** 2 fair
**Rating:** 5
**Confidence:** 3

**Summary:**

This paper introduces Diff-Transfer, a model-based algorithm for transferring manipulation skills by discovering a path from the source task to the target task within the task space. The method views each task as a state-action trajectory, and defines the task using the change of object pose from the initial state to the goal state. Consequently, a path of tasks can define a transfer from the source task to the target task. The transfer can then be solved by optimizing the action sequence based on the gradient from a differentiable simulation. In the end, Q-learning is used for path planning on the task space to find the sub-tasks.

**Strengths:**

* The paper is well-written. I especially found the idea of representing a transfer as a sequence of sub-tasks interesting.
* The experiments in four different tasks demonstrate that the proposed method significantly outperforms other baselines.

**Weaknesses:**

* My main concern of the paper is the range of transfer tasks that the method is able to solve. Primarily, the method seems constrained to solving transfer tasks where the pose of the objects in the scene is changing. In my opinion, this is a relatively easy transfer task, as many standard robot learning tasks would have randomized object initial poses.
* All experimental environments manipulate a single object.

**Questions:**

1. Is it possible for the proposed method to solve other types of transfer tasks than different object poses, like different object shapes or different manipulation goals?
2. I think the problems that this paper is solving are similar to learning an SE(2) or SE(3) equivariant policy (e.g., [A, B]). I am curious about the author's comment on how the proposed method compares to equivariant policy learning, and if it is possible to leverage symmetry in the network to implement direct generalization/transfer.
3. Will the problem space significantly increase if there are multiple objects in the scene?

[A] Wang, Dian, Robin Walters, and Robert Platt. "SO(2)-Equivariant Reinforcement Learning." ICLR, 2021
[B] Simeonov, Anthony, et al. "Neural descriptor fields: SE(3)-equivariant object representations for manipulation." ICRA, 2022.

---

### Official Review · Reviewer_zf8W · 2023-11-01

**Soundness:** 2 fair
**Presentation:** 3 good
**Contribution:** 2 fair
**Rating:** 5
**Confidence:** 2

**Summary:**

The paper proposes a framework (Diff-Transfer) for efficiently transferring robotic manipulation skills between related tasks. The key idea is to create a smooth path of sub-tasks connecting the source and target tasks. At each step, actions are adapted from one sub-task to the next using gradients from a differentiable physics simulator. Experiments on simulated articulated tasks i.e., opening door, opening drawer, closing a grill, and changing a clock demonstrate the effectiveness of Diff-Transfer.

**Strengths:**

1. The paper is generally well-written, with additional video and clear figures.
2. A reasonable approach for robotic transfer learning from one manipulation task to a similar task.
3. Leveraging the differentiable simulation for pose transfer and object shape transfer.

**Weaknesses:**

1. A concern is that the source task and the target task definitions seem simple for robotic manipulation.
2. The introduction and the implementation (experiments) are not really matched - as they claim the task transfer from e.g., pushing to e.g., reorientation, but Fig. 2 shows only a new initial pose of the object. In this case, I wonder if it is only a new configuration of the same task or a new task.

**Questions:**

1. I think some claims in the paper are too strong - e.g., 'revolutionize the paradigm of robotic manipulation skill acquisition'.
2. Fig.3 seems not very informative - the different Q-value plots look similar and the explanation in the paper is unclear.

---

### Official Review · Reviewer_vf9D · 2023-11-02

**Soundness:** 2 fair
**Presentation:** 3 good
**Contribution:** 2 fair
**Rating:** 3
**Confidence:** 3

**Summary:**

This paper proposes an approach to transfer and adapt robotic skills to unseen tasks. The method decomposes the ‘transfer gap’ between a source and target task into a sequence of intermediary tasks, and employ a differentiable simulator as a model by which to determine how to transform the actions along the sequence of intermediary tasks.

The paper addresses an interesting problem and the theoretical development is clear, but I have concerns about the assumptions and applicability of the problem setting, and also feel that the experimental justification is lacking.

**Strengths:**

- It is an interesting problem setting
- The theoretical development seems sound

**Weaknesses:**

- The experimental evaluation needs to be more thorough to be compelling
- The assumptions seem too strong to be useful outside of simpler task settings; and there is insufficient evidence from the experiments to suggest otherwise.

**Questions:**

**Problem setting and assumptions**

The proposed method assumes that there is a successful solution providing actions for the source task, and that the difference in the actions between two adjacent intermediary tasks is small throughout the sequence. The TransferStep algorithm adapts from one task to the next by iteratively (i) rolling out the current actions in the environment; (ii) computing the error in the object displacement; (iii) updating the actions by propagating gradients through the rollout.

Assuming I have understood this correctly, I have a few questions and comments:
1) How robust is the approach to different kinds of transfer gap? Transferring from a pushing task to another (or opening/closing different amounts, or moving an object to a different configuration) may be reasonable, but what about more challenging cases, where the behaviour or affordance might change, or the task might reuse multiple existing skills? Eg. placing an object → inserting an object somewhere else, reaching and grasping an object → lifting an object.
2) If the method is not intended to transfer to different behaviours (only different object displacements), why not just perform planning on the target task directly? TransferStep itself appears to be a form of trajectory optimisation at each step, and at least for the tasks presented in the experiments (which are all relatively simple single-stage manipulation tasks with the object pose changing), I would expect both source and target task to be solvable by planning with the differentiable simulator.

**Experiments**

The authors claim a “rigorous experimental framework” and an “exhaustive evaluation”, but the only evaluation is with 4 source/target tasks and weaker baselines (see my comment below), with a single ablation for the different planning methods used to determine intermediate goals.

The comparison to baselines is not compelling in my opinion, as I would expect them to not succeed in the applied setting.
For example, MAML is a meta-learning approach that learns a good parameter initialisation for adaptation by learning-to-adapt on a very large distribution of tasks. As such, I don’t find it surprising that it fails when training on 4 source tasks (I would expect it to only be useful here if trained on many variants of the same task and adapted to another variant). It feels like the wrong fit for a baseline, and something like finetuning via RL or planning would make more sense to me.
Similarly the direct transfer approach applies the same TransferStep algorithm but without the intermediary goals - as such it is more of an ablation without the planning stage.

**Minor comments**
Figure 1 is confusing. I think it means “to select the best candidate for the (i+1)th subtask…”. The current wording makes it seem like the output of the q function is something else, which ‘generates’ the task.

**Summary**
While I was intrigued by the theoretical development, I think there unfortunately needs to be more evidence that this method could be useful for challenging tasks, along with more clarity around why the proposed problem setting and solution could be useful versus eg. planning in the target domain.

---

### Official Review · Reviewer_orn2 · 2023-11-10

**Soundness:** 1 poor
**Presentation:** 2 fair
**Contribution:** 1 poor
**Rating:** 1
**Confidence:** 4

**Summary:**

This work discusses the importance of efficient skill acquisition for robots in manipulating objects and presents a framework for improving this process through transfer learning. The authors propose a method where robots can transfer knowledge from one task to another by identifying common properties between different manipulation skills. They suggest using a differentiable physics simulator and a refined Q-learning method for path planning to create a series of intermediate sub-tasks that bridge the source and target tasks, allowing for the transfer of action sequences. The framework is tested using a simulator called Jade on various tasks, showing that it outperforms some of the existing methods. The paper's contributions include a framework for model-based transfer learning, a novel path planning method using Q-learning and differentiable physics simulation, and experiments validating the effectiveness of the proposed framework.

**Strengths:**

A paper aims to propose an interesting new method for transfer learning based on differentiable simulation. Transfer learning is a big an important problem in robotics.

**Weaknesses:**

The information provided about the experimental setup is not detailed enough. There is no information about observation space and about task-specific settings. Tasks are very similar to each other so it’s not very clear from the experiments how much transfer learning actually happens and if the approach will work for more different tasks. The conclusion section is a bit too ambitious given the set of experiments and experimental results provided in the paper.

**Questions:**

1) Could you provide more detail about the experimental setup, observations, action space, and rewards for each of the tasks? How start robot states are randomized?

2) How do you conclude from the experiments that transfer learning actually happens?

---

### Official Review · Reviewer_SmsY · 2023-11-10

**Soundness:** 3 good
**Presentation:** 2 fair
**Contribution:** 2 fair
**Rating:** 3
**Confidence:** 3

**Summary:**

This paper introduces Diff-Transfer, a framework for transfer learning of robot skills using a differentiable physics-based simulation. They propose a path planning method which splits the problem of transfer from source to target task into multiple subtasks. Each subtask is a new sample in goal space which is evaluated by a learned critic network. The action sequence is then provided via the differentiable physics simulation.

**Strengths:**

- Diff-Transfer is able to accomplish the evaluation tasks successfully while the baseline methods mostly fail to complete the tasks
- Requires fewer iterative steps compared to the Direct Transfer baseline
- Provided analysis on other path-planning methods such as linear interpolation and an ablation over the task term in the reward function

**Weaknesses:**

- Wording is overly exaggerated in the conclusion: " ... our pioneering
contributions herald a new era in robotic adaptability ... ". Word choice is a bit flamboyant in multiple places in the writing.
- This paper seems to only be tackle in-distribution task-transfer where typically transfer is thought of as learning task A can help with a completely different task B.
- Additionally, object shape transfer is mentioned as one of the applications, but only object pose transfer is considered in the experiments.
- Reward function seems to be very hand-engineered. How many data points is required to fit the Q-network with the pretraining dataset? Is this dataset hard to collect?
- Is there any comparison with other works that use differentiable physics for task transfer?
- One of the claimed novelty in this work is the path planning algorithm for sampling new subtasks. Can you include more comparisons against other path planning algorithms in classical literature like RRT, A*, sampling-based methods, etc?

Typo and writing comments:

Figure 1: Sub-Task Accomplishment ...

Section 5.2 MAML: repeated the word "application"

Confusing last sentence in Section 5.1.4.

Section 5.3, why is this transfer task considered "innovative"?

**Questions:**

- How scalable is this method to more severe pose changes or object shape changes?
- Why do you mention that this is an RL method when all you are learning is a critic network?

---

### Meta-Review · Area_Chair_JfCx · 2023-12-06

**Metareview:**

(a) Summary
The paper introduces a framework for transfer learning in robotic manipulation tasks. The key idea is to create a smooth path of sub-tasks connecting the source and target tasks. At each step, actions are adapted from one sub-task to the next using gradients from a differentiable physics simulator. Experiments on simulated articulated tasks i.e., opening door, opening drawer, closing a grill, and changing a clock demonstrate the effectiveness of the approach.

(b) Strengths of the Paper:

(+) Effective Skill Transfer in Robotic Manipulation (Reviewers SmsY, PT7k): Diff-Transfer shows good performance in simulation experiments. It successfully completes tasks where baseline methods struggle.

(+) Efficiency in Iterative Steps (Reviewer SmsY): The framework requires fewer iterative steps compared to the Direct Transfer baseline, showcasing its efficiency.

(c) Weaknesses of the Paper:

(-) Exaggerated Claims and Writing Style (Reviewer SmsY): The paper's conclusion and other sections use exaggerated language, detracting from its scientific merit.

(-) Limited Scope in Task Transfer (Reviewer SmsY, PT7k): The paper primarily addresses in-distribution task-transfer, raising questions about its effectiveness in transferring skills across substantially different tasks.

(-) Specificity in Object Shape Transfer (Reviewer SmsY): Although object shape transfer is mentioned, the experiments only demonstrate object pose transfer, limiting the scope of the claimed applications.

(-) Detailed Methodological Clarifications Needed (Reviewers orn2, vf9D): The paper lacks detailed information about experimental setups and assumptions. More clarity is needed on the experimental conditions, such as the observation space, action space, and rewards for each task.

(-) Insufficient Experimental Validation (Reviewer vf9D): The experimental evaluation seems inadequate to support the claims fully. The experiments are limited to a few tasks with weak baselines, lacking a thorough and diverse validation of the proposed method.

(-) Assumptions and Applicability Concerns (Reviewer vf9D): The method's assumptions are considered too strong for broader applicability. There is insufficient evidence to suggest the method's effectiveness outside simpler task settings.

**Justification For Why Not Higher Score:**

There were several weaknesses that were highlighted above that the authors did not provide any rebuttal against

**Justification For Why Not Lower Score:**

N/A

---

### Decision · Program_Chairs · 2024-01-16

Reject